# Rethinking the Roles of Time and Frequency Domains Before Tackling Time Series UDA

## Abstract

In time-series unsupervised domain adaptation (UDA), the adaptation between temporal and frequency domain features has been relatively underexplored. To address this gap, we conduct a comprehensive series of experiments to revisit the roles of these domains in source-free UDA (SFUDA), a branch of the UDA task. Our findings reveal that the temporal domain contains more diverse features, offering higher discriminability, while the frequency domain is more domain-invariant, providing better transferability. Combining the strengths of both domains, we propose TidalFlow, a SFUDA framework that synergistically integrates temporal and frequency domain features. TidalFlow enhances feature extraction and captures subtle, class-specific features without relying on traditional alignment strategies. By utilizing simple hyperparameter adjustments and using frequency embeddings from the source domain as reference points for domain adaptation, TidalFlow achieves nearly a 10% improvement across five benchmark datasets in time-series UDA. This research highlights the unique strengths of both domains and marks a paradigm shift in SFUDA methods, showcasing TidalFlow's robust performance in real-world applications. Code is available at the anonymous link: https://anonymous.4open.science/r/TidalFlow-42B0/.

## 1 Introduction

Time series datasets showcase the prowess of neural networks Ravuri et al. (2021); Lundberg et al. (2018), but their vulnerability to domain shifts poses deployment challenges Singhal et al. (2023); Painblanc et al. (2023); Zhang et al. (2021). These shifts, stemming from nuanced differences in test distributions, hinder model generalization Koh et al. (2021); Luo et al. (2018); Zhang et al. (2013). Addressing this, domain adaptation (DA) techniques, such as leveraging unlabeled data Garg et al. (2021); Ganin et al. (2016), emerge as essential to ensure robust model performance in real-world scenarios. In addition, DA for time series is even more difficult Wilson & Cook (2020); Ozyurt et al. (2023); He et al. (2023), as it has to deal with both the domain discrepancy and the temporal dynamics that may cause feature shift and label shift.

Unsupervised Domain Adaptation (UDA) is pivotal for enhancing the generalization of machine learning models, aiming to train a model on a labeled source domain that can effectively perform on a related yet unlabeled target domain Garg et al. (2021); Ganin et al. (2016). While UDA methods have flourished in computer vision Huo et al. (2022); Tang et al. (2021); Pan et al. (2020); Tzeng et al. (2019), their application to time series, though feasible with feature extractor adjustments, often falls short in fully harnessing time-series properties. In the domain of time series, a limited number of works have explicitly addressed UDA, they most focus on temporal information. Even when the frequency domain is considered, it is typically combined with temporal features and treated as general information during training.

To clarify the characteristics of the time and frequency domains, this research conducted a series of experiments leading to the following conclusions: **the temporal domain provides broader information with stronger classification discriminability, while the frequency domain, though simpler, offers more domain-invariant features that serve as reference points between the source and target domains** (Section 3).

Our research integrates the strengths of both the temporal and frequency domains, moving beyond the prior focus on "how to align two inconsistent distributions" to explore "**how to identify features**

**that represent classes across domains."** The difference lies in that the former approach pays little attention to the features extracted by the model, focusing instead on alignment methods and classifier performance. This overemphasis on alignment leads to overly sensitive and inflexible classifiers, particularly when dealing with data with large domain gaps or longer time series. The latter approach avoids these pitfalls by enabling the model to utilize class-representative features early in training, ensuring more robust performance.

We propose TidalFlow, a simple framework for SFUDA in time series that leverages both temporal and frequency domain characteristics to achieve strong performance. Our model integrates information from both domains to capture subtle, class-specific features, enhancing feature extraction. By focusing on the domain-invariant properties of the frequency domain, we use a frequency embeddings table from the source domain as reference points, along with simple hyperparameter adjustments, to enable the model to find the most suitable embeddings for target domain data during adaptation, ultimately assigning the appropriate class labels. This straightforward training framework showcases the complementary strengths of the temporal and frequency domains, resulting in exceptional performance across five different real-world datasets.

**Contributions:**

1. Through a series of experiments, we revisited the key components of the temporal and frequency domains and concluded that the temporal domain provides richer information with better discriminability. In contrast, the frequency domain, due to its inherent properties, offers more structural features that are domain-agnostic between source and target domains, resulting in superior transferability.

2. We introduce TidalFlow, a model architecture based on VQ-VAE specifically designed for SFUDA in time series. This framework strategically integrates information from both domains using a frequency embedding table to effectively determine optimal embeddings for target domain data.

3. TidalFlow exhibits nearly 10% significant improvement across five benchmark datasets for time-series UDA, underscoring its competitive edge in this field.

## 2 RELATED WORK

### 2.1 UNSUPERVISED DOMAIN ADAPTATION

Unsupervised domain adaptation (UDA) involves utilizing labeled data from a source domain to predict labels for an unlabeled target domain. The primary objective of UDA methods is to minimize domain discrepancy, thereby reducing the lower bound of target error. Existing UDA approaches can be broadly categorized into three groups: (1) Metric-based methods, like DDC (Tzeng et al., 2019), Deep CORAL (Sun & Saenko, 2016), DeepJDOT (Damodaran et al., 2018), HoMM (Chen et al., 2020), and MMDA (Rahman et al., 2020), minimize domain discrepancy by imposing restrictions using a distance metric (e.g., maximum mean discrepancy). (2) Adversarial-based methods employ domain discriminator networks, such as DANN (Ganin et al., 2016), CDAN (Long et al., 2018), and DIRT-T (Shu et al., 2018), to enforce the feature extractor in learning domain-invariant representations. (3) Contrastive methods reduce domain discrepancy through a contrastive loss, aligning embeddings of source and target samples of the same class. Pseudo-labels, generated by clustering algorithms, are used for target samples, as their actual labels are unknown. Examples include CAN (Kang et al., 2019), CLDA (Singh, 2021), and IDCo (Zhang et al., 2023). While UDA has been extensively explored in computer vision, limited research has been conducted on UDA for time-series data.

### 2.2 TIME-SERIES UNSUPERVISED DOMAIN ADAPTATION

Despite successes in computer vision, there has been a notable gap in research focusing on adaptation methods tailored for time-series data. Few methods have been specifically crafted for time-series domain adaptation. (1) Adversarial training for time-series UDA involves using adversarial methods to learn domain-invariant temporal relationships, such as VRADA (Purushotham et al., 2017), and CoDATS (Wilson et al., 2020). (2) Statistical divergence methods for time-series UDA focus on aligning the statistical properties of source and target domains. Examples include SASA (Cai et al.,

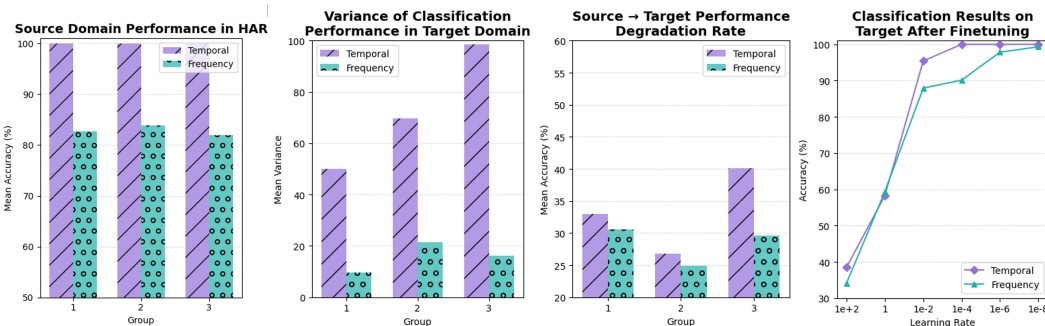

Figure 1: We randomly selected 15 source $\mapsto$ target pairs from the HAR dataset and divided them into three groups for analysis, focusing on the following metrics: (a) mean accuracy in the source domain, (b) mean variance in classification performance within the target domain, and (c) the performance degradation rate when testing the source domain pre-trained model on the target domain. Additionally, we evaluated the impact of hyperparameters on transferability in both the time and frequency domains by assessing (d) the mean accuracy after fine-tuning with different learning rates.

2021), AdvSKM (Liu & Xue, 2021a) and Ott et al. (2022). (3) Self-supervision methods for time-series UDA extract domain-invariant and domain-specific features. DAF (Jin et al., 2022) uses a shared attention module with a reconstruction task. Contrastive methods like (Ozyurt et al., 2023), CoTMix (Eldele et al., 2023), and CALDA (Wilson et al., 2023) use augmentations to enhance prediction. RAINCOAT (He et al., 2023) addresses feature and label shifts by aligning them across domains. Despite their potential, they rely on access to source data, which may not always be feasible due to privacy concerns.

A more practical method in the real world is the SFUDA task, which can perform domain adaptation without source data and target labels. Liang et al. (2020) freezes the source model's classifier and uses information maximization and self-supervised pseudo-labeling to align target domain representations to the source hypothesis. And Ragab et al. (2023b) captures temporal information through random masking and a temporal imputer to ensure temporal consistency between source and target features during adaptation. TemSR (Wang et al., 2024) transfers temporal dependencies without requiring source-specific designs by leveraging masking, recovery, and optimization to generate a source-like distribution for adaptation. However, these methods have not taken full advantage of both time and frequency domain properties in addressing the UDA problem.

### 2.3 VECTOR QUANTISED VARIATIONAL AUTOENCODER (VQ-VAE)

Conceptualized as a communication system, the VQ-VAE (Van Den Oord et al., 2017) model comprises an encoder and a decoder. The encoder involves a non-linear mapping from the input space to a vector, which is then quantized by determining its nearest prototype vector in a shared codebook. The quantized vector, essentially the index of the closest prototype vector, is transmitted to the decoder. Despite the potential loss, the decoder maps these indices back to their corresponding vectors in the codebook, reconstructing the data through another non-linear function. Learning involves back-propagating the gradient of the reconstruction error through the decoder and to the encoder, utilizing the straight-through gradient estimator.

A key benefit of VQ-VAE is its discrete representation, which proves useful in obtaining effective features. In UDA, data distribution from the target domain is indirectly captured through self-supervised learning. Notably, VQ-VAE is less susceptible to model degeneration issues, enabling it to effectively capture both temporal and frequency domain information during adaptation without the associated concerns.

## 3 PROBLEM FORMULATIONS

### 3.1 SCENARIO DESCRIPTION

We are given two distributions of time-series data: one from the source domain $D_s$ and the other from the target domain $D_t$. In this setup, define labeled *i.i.d.* samples from the source domain as $S = \{(\mathbf{x}_i^s, \mathbf{y}_i^s)\}_{i=1}^{N_s} \sim D_s$, where $\mathbf{x}_i^s$ represents a sample from the source domain, $\mathbf{y}_i^s \in \{1, ..., H\}$, where $H$ is the number of classes, and $\mathbf{y}_i^s$ denotes the label for the corresponding sample, and $N_s$ denotes the total number of *i.i.d.* samples in the source domain. Conversely, consider **unlabeled** *i.i.d.* samples from the target domain denoted by $T = \{\mathbf{x}_i^t\}_{i=1}^{N_t} \sim D_t$. Here, $\mathbf{x}_i^t$ denotes an individual sample from the target domain, and $N_t$ represents the total number of *i.i.d.* samples collected from the target domain. Furthermore, each $\mathbf{x}_i$, whether originating from $D_s$ or $D_t$, constitutes a sample of a multivariate time series denoted by $\mathbf{x}_i = \{\mathbf{x}_{i,t}\}_{t=1}^{L} \in \mathbb{R}^{M \times L}$, where $L$ represents the number of time steps, and $\mathbf{x}_{i,t} \in \mathbb{R}^M$ signifies $M$ observations for the respective time step.

Our objective is to establish an embedding table through UDA on the source samples $S$, enabling effective generalization on the target samples $T$. Notably, in the provided time series datasets for $D_s$ and $D_t$, where the label sets are identical $C_s = C_t$, the target labels $y_t$ **are not** available during the training phase.

The aforementioned scenario is practically relevant across various applications Feng et al. (2023); Ramponi & Plank (2020); Zhang et al. (2018), whether in machine faulty detection Lessmeier et al. (2016), predicting the four sleep stages using EEG signals Goldberger et al. (2000), or recognizing human activity Stisen et al. (2015); Anguita et al. (2013); Kwapisz et al. (2011) through signals from wearable devices. The differences in machines, environments, and individuals can easily lead to significant domain shifts in the datasets. Therefore, to ensure accurate predictions and generalization, it is often necessary to adapt and apply deep learning models trained in one domain $S$ to another domain $T$.

### 3.2 PRELIMINARY STUDY

We design a series of experiments on both the temporal and frequency domains. To minimize model influence, we follow prior research (Liu & Xue, 2021b; Cheng et al., 2024) by constructing a 3-layer CNN as a temporal feature extractor and a frequency feature extractor that combines a fast Fourier transform with a 1-layer linear network. Both are followed by a 1-layer linear classifier for simplicity.

The key question we explore is: ***What kind of feature information do the temporal and frequency domains provide?*** We pre-train three models on the source domain until ~~until~~ they converge and observe their performance on the target domain. During the temporal model experiments, we observe a noteworthy phenomenon: despite achieving nearly 100% accuracy in the source domain (Fig. 1(a)) with different model parameters, the performance on the target domain exhibits considerable fluctuation. As shown in Fig. 1(b), the performance variance of the three temporal models is larger than that of the frequency models. A t-test confirms a statistically significant difference in performance variance between the temporal and frequency models (p-value = 0.0133).

**Transferability.** When we examine transferability, Fig. 1(c) shows that the temporal models experience a more significant performance drop, with a statistically significant difference from the frequency models (p-value = 0.0491). We hypothesize that this is because the temporal domain contains a wider variety of information, enabling the model to classify based on multiple dimensions. Nevertheless, this diverse information also includes more features specific to the source domain or confounders, meaning that when domain shifting occurs, the model's focus may no longer be on the relevant class features of the target domain, resulting in poorer transferability.

In contrast, the frequency domain, after undergoing Fourier transformation, filters out much of the extraneous information, such as signal start and end points or noise, resulting in fewer feature dimensions. However, this allows the frequency models to focus more on the overall structure of the information, making them more domain-invariant. Fig. 1(c) supports this, showing that although the frequency models do not perform as well as the temporal models in source domain classification, their transferability is superior.

This raises another concern: ***Is the frequency domain truly more domain-invariant?*** To investigate, we design another experiment where we only adjust the extent of feature updates (here, we choose to adjust the learning rate) during the fine-tuning phase. Our assumption is that if merely tweaking the learning rate significantly improves model performance, it indicates that the frequency domain contains domain-agnostic features that are specific to each class of data, rather than just irrelevant features that do not contribute to the model's effectiveness.

As shown in Fig. 1(d), the frequency models require a very small learning rate to fine-tune correctly. Larger learning rates prevent the frequency models from converging to the optimal point. Interestingly, the temporal models are much less sensitive to hyperparameter adjustments compared to the frequency models. In Fig. 1(d), despite averaging accuracy across 15 source $\mapsto$ target experiments, the temporal models fine-tune to 100% accuracy across learning rates ranging from $1 \times 10^{-4}$ to $1 \times 10^{-8}$. This could be explained by the high feature diversity in the temporal domain, allowing different model parameters to reach optimal solutions depending on the learning rate. Meanwhile, the frequency models retain robust domain-invariant features between source and target domains, making them better suited to fine-tuning with smaller steps.

**Empirical insights.** The analysis reveals two key insights regarding time-series domain adaptation: (1) the time domain excels at classification, but its transferability is hindered by an excess of confounding factors, and (2) the frequency domain, though containing more uniform and less diverse information, offers better domain-invariant features, leading to stronger transferability. Based on these observations, we design a simple model framework that leverages the rich features of the time domain while using the frequency domain as a reference point to bridge the source and target domains. Our experimental results demonstrate that combining the strengths of both domains yields improved performance.

## 4 OUR APPROACH

Next, we present the architecture of TidalFlow, which consists of three modules: a dual-stream encoder $G$, a hierarchical embedding table (HET), a 1-layer linear classifier for training, and a decoder $U$ for adaptation. Section 4.1 introduces an encoder network $G$, which extracts both temporal and frequential features from the input. Section 4.2 introduces how the hierarchical embedding table (HET) be initialized and how it works during different phases. Section 4.3 introduces the voting mechanism after the nearest-neighbor algorithm in the inference phase. We follow the framework as VQ-VAE (Van Den Oord et al., 2017) that uses the selected embeddings as input into the decoder $U$. Section 4.4 outlines the objective functions during the training and adaptation phases and provides an overview of TidalFlow.

### 4.1 DUAL-STREAM ENCODER $G$

$G$ encodes both time and frequency representations, and the source temporal and frequential features are denoted as $\mathbf{z}_{temp,i}^s$ and $\mathbf{z}_{freq,i}^s$, while the target features are denoted as $\mathbf{z}_{temp,i}^t$ and $\mathbf{z}_{freq,i}^t$. We will employ the simplified terms $\mathbf{z}_{temp}$ and $\mathbf{z}_{freq}$ to collectively represent features from both $D_s$ and $D_t$ in the subsequent explanations. By including frequency information, the encoder enhances its ability to adapt across domains by potentially identifying common features. The encoder parameterizes a posterior distribution $q(\mathbf{z}|\mathbf{x})$ over the latent variables $\mathbf{z}_{temp}$ and $\mathbf{z}_{freq}$ based on the input. This posterior captures relationships between the input and latent representations, informed by both temporal and frequency patterns extracted from the input, the following:

$$G(\mathbf{x}) = \text{Concat}[\mathbf{z}_{temp}(\mathbf{x}), \mathbf{z}_{freq}(\mathbf{x})], \quad \forall \mathbf{x} \in D, \tag{1}$$

where $D$ is either $D_s$ or $D_t$, and Concat is the abbreviation of concatenation.

### 4.2 HIERARCHICAL EMBEDDING TABLE (HET)

**Initialization.** We introduce a 2-layer top-down embedding table and the initial layer is organized based on task labels, consisting of $H$ categories. The subsequent layer of the hierarchical embedding table comprises independent latent embedding spaces for each $\mathbf{e}_h$, denoted as $\mathbf{e}_h \in R^{K \times \Psi}$, where $K$

Figure 2: The TidalFlow framework. (a) During training, input data $\mathbf{x}_i^s$ undergoes processing through the Dual-Stream Encoder $G$ to generate a temporal and frequency combined feature representation $\mathbf{z}_e^s$. Representative embeddings are retrieved from HET based on input labels, and the classifier distinguishes between the categories. The function $\rho$ is employed for finding the nearest embedding for $\mathbf{z}_e$ in HET. (b) In adaptation, TidalFlow adjusts embeddings in HET using frequency reference points to tackle domain shifts through a reconstruction task with the decoder $U$. (c) During inference, a voting mechanism ranks similarities between embeddings and $\mathbf{z}_e^t$ to enhance classification.

represents the number of the discrete latent variables of each category and $\Psi$ is the dimensionality of each embedding vector. To sum up, there are $H \times K$ embeddings in the hierarchical embedding table and we initialize the embeddings by uniform distribution.

**Training phase.** We perform a nearest neighbor search in the whole embedding space, focusing on the category in the source domain that corresponds to the input $\mathbf{x}$ as outlined in Eq. 2. The probabilities of the posterior categorical distribution $q(G(\mathbf{x})|\mathbf{x})$ are defined as one-hot encoded, following:

$$q(G(\mathbf{x}) = k|\mathbf{x}) = \begin{cases} 1 & \text{for } k = \arg\min_j \|G(\mathbf{x}) - \mathbf{e}_{h,j}\|_2, \\ 0 & \text{otherwise} \end{cases}, \tag{2}$$

where $h$ denoted to the same category as $\mathbf{x}$ and $j$ is the candidates of the category $h$.

**Adaptation phase.** Due to the lack of labels in $D_t$, the model cannot search for the most similar embeddings within the respective categories. Therefore, we take advantage of the distinctive characteristics of the frequency domain and partially freeze the frequency blocks of HET. This deliberate constraint, achieved through significantly different learning rates, establishes a clear reference point for the encoded latent representations. Consequently, both the time and frequency modules can efficiently navigate the gradient map, leading to the identification of optimal solutions with appropriately adjusted update steps. Accordingly, we can modify Eq. 2 to be agnostic to the category $h$:

$$q(G(\mathbf{x}) = k|\mathbf{x}) = \begin{cases} 1 & \text{for } k = \arg\min_j \|G(\mathbf{x}) - \mathbf{e}_j\|_2, \\ 0 & \text{otherwise} \end{cases}, \tag{3}$$

where $j$ is the embeddings of HET and there is no category $h$ in this equation.

### 4.3 VOTING MECHANISM

After the training and adaptation phases, the embeddings in HET have formed $H$ distinctive clusters. This implies that, while the embedding in HET is discrete, the majority possess representative features specific to their respective categories $h$. Subsequently, we employ a nearest-neighbor algorithm to determine the top $K$ categories (where $K$=5) represented by the embeddings. Through a voting

mechanism, we ascertain the category to which the input data should belong. This enhances the robustness of TidalFlow. The algorithm of the voting mechanism can be seen in the Appendix A.

## 4.4 OBJECTIVE FUNCTIONS

In TidalFlow, we utilize three types of objective functions during the training phase: (1) classification loss, (2) dissimilarity loss, and (3) feature-embedding consistency loss. While there are two types of objective functions during the adaptation phase: (1) reconstruction loss and (2) feature-embedding consistency loss.

**Classification loss** $\mathcal{L}_{\text{CE}}$. We utilize cross-entropy loss as the loss function for our classification task during training.

**Dissimilarity loss** $\mathcal{L}_{\text{D}}$. This objective function is designed to prevent the model from generating nearly identical embeddings among categories during the training phase. To achieve this, we identify the closest embedding to $\mathbf{z}_{freq}$ from all embeddings in the frequency block, which is more domain-agnostic than temporal features and does not belong to the same category as $\mathbf{y}_i^s$. The repulsive effect is introduced by calculating the dissimilarity loss. It is worth noting that, while TidalFlow searches for the closest representative in the embedding table within the same category as $\mathbf{x}_i^s$, this approach may result in the model learning a common feature across all categories, neglecting latent features that distinguish between different categories. To address this, we utilize the following equation (Eq. 4) to guide the model explicitly in generating a better latent representation.

$$\mathcal{L}_{\text{D}} = 1 - \|\text{sg}[\mathbf{e}_{freq[h \neq \mathbf{y}]}] - \mathbf{z}_{freq}\|_2^2 \tag{4}$$

where $\mathbf{e}_{freq[h \neq \mathbf{y}]}$ is the chosen embedding from the frequency block on the hierarchical embedding table, and its category $h$ cannot be the same label of the input data $\mathbf{x}_i^s$. Additionally, sg($\cdot$) represents the stop-gradient operator, which functions as an identity during forward computation and possesses zero partial derivatives.

**Feature-embedding consistency loss** $\mathcal{L}_{\text{A}}$. Taking inspiration from VQ-VAE (Van Den Oord et al., 2017), TidalFlow incorporates vector quantization algorithms, guiding the embedding encoder outputs towards proximity through L2 error, thus effectively learning the embedding space. The hierarchical structure of the embedding table, divided into temporal and frequency blocks, assigns each block to handle specific features. Consequently, they do not share the same optimizer but are updated independently. Additionally, to address a concern highlighted by VQ-VAE about the lack of dimensionality constraints on the embedding space, which could potentially lead to uncontrolled growth, TidalFlow adjusts the weight of this constraint to $\alpha$ and $\beta$ for both temporal and frequency blocks. The objective function is expressed as:

$$
\begin{aligned}
\mathcal{L}_{\text{A}} = \alpha \overbrace{(\|\text{sg}[\mathbf{e}_{freq}] - \mathbf{z}_{freq}\|_2^2) + \|\mathbf{e}_{freq} - \text{sg}[\mathbf{z}_{freq}]\|_2^2}^{\text{Frequency block}} \\
+ \beta \underbrace{(\|\text{sg}[\mathbf{e}_{temp}] - \mathbf{z}_{temp}\|_2^2) + \|\mathbf{e}_{temp} - \text{sg}[\mathbf{z}_{temp}]\|_2^2}_{\text{Temporal block}}
\end{aligned}
\tag{5}
$$

**Reconstruction loss** $\mathcal{L}_{\text{MSE}}$. During the adaptation phase, since the representative chosen from the hierarchical embedding table does not provide the model with a real gradient, we employ the straight-through estimator (Van Den Oord et al., 2017). This allows us to directly pass the gradient generated by the decoder back to the encoder. We opt not to use the subgradient through the quantization operation, as VQ-VAE has demonstrated that a simple estimator can achieve effective training outcomes. As the output representation of the encoder and the input to the decoder exist in the same $D$-dimensional space, the gradients carry valuable information on how the encoder needs to adjust its output to minimize the reconstruction loss.

**Overview of TidalFlow.** During training, we employ the classification loss for our classification task. The total loss function is defined with three components in the objective function, as outlined below:

$$\mathcal{L}_{\text{training}} = \mathcal{L}_{\text{CE}} + \mathcal{L}_{\text{A}} + \mathcal{L}_{\text{D}}. \tag{6}$$

During adaptation, we replace the classification task with a reconstruction task, which leads us to modify our objective function as shown in Eq. 7. This design enables TidalFlow to outperform other time-series UDA methods. Last but not least, an overview algorithm of TidalFlow is in Appendix A.

$$\mathcal{L}_{\text{adaptation}} = \mathcal{L}_{\text{MSE}} + \mathcal{L}_{\text{A}}. \tag{7}$$

## 5 Experiments

### 5.1 Experimental setup

**Datasets.** We employ a comprehensive evaluation strategy, consisting of two main aspects. First, extensive experiments are conducted using five well-established benchmark datasets in UDA tasks, from three distinct problem types: (1) Human Activity Recognition: HAR (Anguita et al., 2013), HHAR (Stisen et al., 2015), WISDM (Kwapisz et al., 2011); (2) Sleep Stage Classification: Sleep-EDF (Goldberger et al., 2000); (3) Machine Fault Diagnosis: MFD (Lessmeier et al., 2016). In human activity recognition datasets, we treat sensor measurements from each participant as distinct domains. To ensure robust assessment, we randomly select 10 source-target domain pairs for evaluation, a methodology widely adopted in previous works on UDA in time-series research (He et al., 2023; Ozyurt et al., 2023; Cai et al., 2021; Wilson et al., 2020). For the sleep stage classification task, following the approach of (Ragab et al., 2023a), we utilize the Sleep-EDF dataset, comprising EEG readings from 20 healthy subjects, and we specifically choose EEG in alignment with previous studies (Eldele et al., 2021). The machine fault diagnosis dataset has been collected under four different operating conditions, and we treat them as separate domains. In contrast to datasets used for human activity recognition being multi-variate, the data used in Sleep-EDF and MFD consist of a single univariate channel following previous works. (He et al., 2023; Ragab et al., 2023a) Further details on datasets are given in Appendix B.

**Baselines.** We evaluate nine domain adaptation methods, including general UDA approaches: deep correlation alignment (Deep Coral) (Sun & Saenko, 2016), decision boundary iterative refinement training with a teacher (DIRT-T) (Shu et al., 2018), HoMM (Chen et al., 2020), and CDAN (Long et al., 2018). Additionally, we include four UDA methods specifically designed for time series: CoDATS (Wilson et al., 2020), adversarial frequency kernel matching for unsupervised time-series domain adaptation (AdvSKM) (Liu & Xue, 2021a), contrastive learning for unsupervised domain adaptation of time series (CLUDA) (Ozyurt et al., 2023), and RAINCOAT (He et al., 2023). As a baseline, we also consider source-domain-only training (no transfer) using the time-frequency encoder as RAINCOAT (He et al., 2023) and a 1-layer classifier.

**Evaluation.** We present accuracy and macro-F1 scores computed based on the target test datasets. In the experiment, we assign the values of 1 to both parameters $\alpha$ and $\beta$, treating the time domain and frequency blocks as equally important. More hyperparameter settings can be seen in Appendix D.

### 5.2 Results

#### 5.2.1 Classification performance on DA benchmark datasets

In Fig. 3, the average accuracy of each method is presented across 10 sources $\mapsto$ target domain pairs on the HAR, HHAR, WISDM, Sleep-EDF, and MFD datasets. On the HAR dataset, our model surpasses the best baseline accuracy achieved by RAINCOAT by 1.93% (0.844 vs. 0.828). For the HHAR dataset, our model outperforms the best baseline accuracy of CLUDA by 5.5% (0.624 vs. 0.569). In the case of the WISDM dataset, our model excels by surpassing the best baseline accuracy of RAINCOAT by 21.34% (0.688 vs. 0.567). Moving on to the Sleep-EDF dataset, our model exceeds the best baseline accuracy of DIRT-T by 9.1% (0.779 vs. 0.714). Similarly, on the MFD dataset, our model beats the best baseline accuracy of DIRT-T by 11.73% (0.819 vs. 0.733). Despite our model's simplicity compared to state-of-the-art methods, it achieves the highest scores across five different datasets. The Appendix C contains a detailed compilation of UDA results for each source $\mapsto$ target pair, accompanied by Macro-F1 scores, which further support our conclusions.

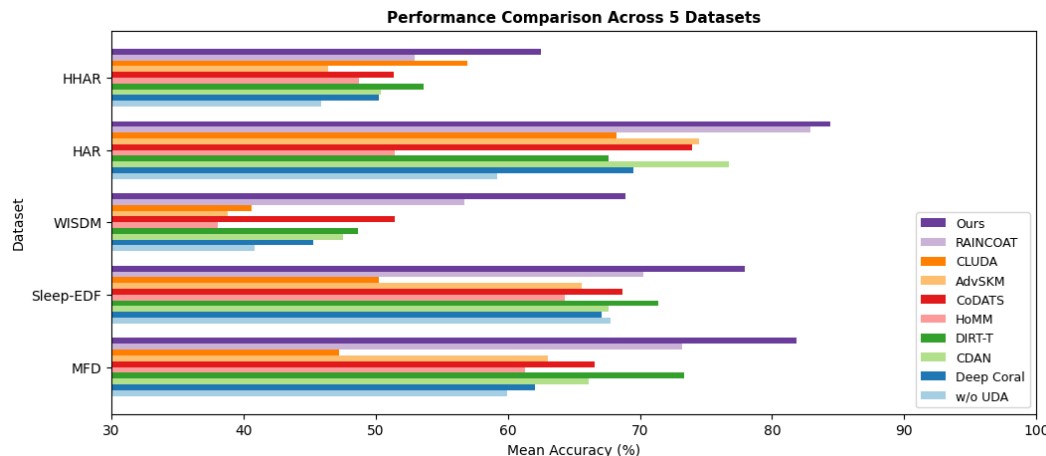

Figure 3: Average performance of multiple DA methods across 5 real-world time-series datasets. TidalFlow consistently outperforms all other methods in accuracy on test sets drawn from the target domain dataset.

### 5.2.2 DIFFERENT FREQUENCY AND TEMPORAL BLOCK LEARNING RATES

We further analyze the impact of different learning rates for the temporal and frequency blocks of TidalFlow during the adaptation phase. We conduct experiments using the MFD and Sleep-EDF datasets due to their large data volumes, which make performance differences more pronounced, as shown in Fig. 4. We discover some valuable findings:

1. When the learning rate for the frequency block is smaller, TidalFlow's adaptability improves. This trend aligns with the observations of our insights in Section 3.

2. When the learning rate for the temporal block is larger, the model's performance deteriorates. We speculate that this is due to the interaction between the encoder and the HET within TidalFlow architecture. Specifically, when the learning rates of the temporal and frequency blocks differ by four orders of magnitude, it indirectly hinders the adjustment range of one of the blocks through the encoder.

Therefore, we recommend setting the learning rates of the temporal and frequency blocks to the same value during the adaptation phase for optimal performance.

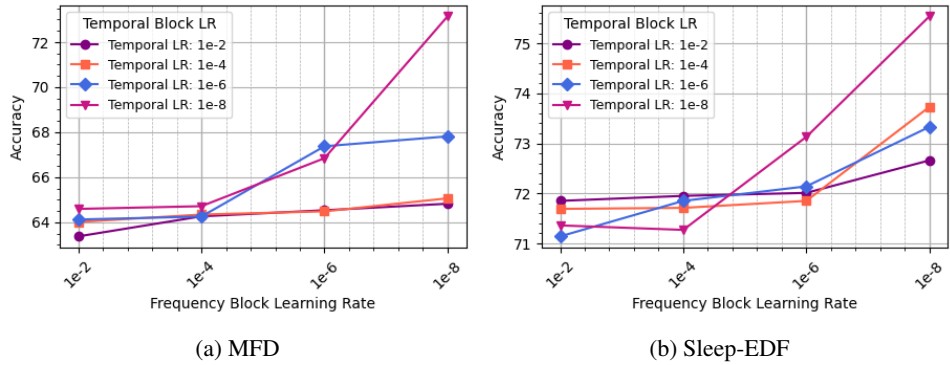

(a) MFD  (b) Sleep-EDF

Figure 4: Accuracy for Different Frequency and Temporal Block Learning Rates in (a) and (b) Dataset.

### 5.3 EMBEDDINGS IN HET AFTER TRAINING PHASE

Table 1: The ablation study of TidalFlow, where performance is measured in terms of accuracy (%).

| ELEMENT OF OUR MODEL | | | MFD DATASET | | | |
|---|---|---|---|---|---|---|
| FREQUENCY BLOCK | $\mathcal{L}_D$ | VOTING | $1 \mapsto 3$ | $2 \mapsto 1$ | $3 \mapsto 2$ | AVG |
| (A) ✓ | | | 83.94 | 80.23 | 77.81 | 80.66 |
| (B) | | ✓ | 58.36 | 65.45 | 69.10 | 64.30 |
| (C) ✓ | ✓ | | 87.25 | 86.08 | 84.19 | 85.84 |
| (D) ✓ | | ✓ | 83.81 | 85.77 | 82.59 | 84.06 |
| (E) ✓ | ✓ | ✓ | **99.84** | **91.71** | **87.22** | **92.92** |

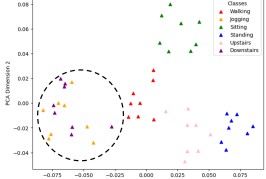

(a) Temporal block

To further understand why TidalFlow succeeds in UDA tasks, we utilize principle component analysis (PCA) to visualize the embeddings in 2D and observe the distribution of embeddings from the temporal block and the frequency block. Fig. 5 shows that even though we initialize the embeddings of both blocks uniformly in the HET, the trained embeddings of the temporal block do not cluster as effectively as those of the frequency block.

This may be due to the higher diversity and complexity of features in the time domain. These features include not only class-specific characteristics but also information such as confounders. In contrast, the frequency block contains more uniform and less diverse information, which allows it to learn the key features of the category more effectively during training. As a result, it demonstrates better clustering performance in the PCA visualization (Fig. 5(b)), aligning with the findings from earlier experiments in Section 3.

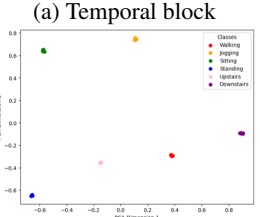

(b) Frequency block

Figure 5: PCA visualization of (a) temporal features and (b) frequency features in HET from WISDM dataset.

### 5.3.1 ABLATION STUDY

To better understand the impact of different components in TidalFlow, we conducted ablation experiments on three key elements: the frequency block, dissimilarity loss $\mathcal{L}_D$, and the voting mechanism, employing five different configurations (Table 1). Given that TidalFlow relies on the frequency block as a reference point, experiments without the frequency block (Table 1 row (B)) exclusively utilized the temporal block for adaptation. Notably, experimental setups without the frequency block and with $\mathcal{L}_D$ were not feasible, considering that $\mathcal{L}_D$ is computed based on the frequency embedding table.

During the inference phase, TidalFlow utilizes a voting technique. In the ablation experiment settings, we adjust the 'without voting' configuration to directly select the category of the most similar embedding as the final prediction.

The results reveal that the absence of both the frequency block and $\mathcal{L}_D$ (Table 1 row (B)) leads to the poorest performance. Conversely, having only the frequency block (Table 1 row (A)) significantly improves classification accuracy. This underscores the argument presented in our preliminary study that the frequency domain's domain-invariant properties between source and target domains enable TidalFlow to generate distinct feature distributions for each category during training. The use of the well-learned frequency embedding table as a robust reference guides the classification of target domain data into the correct categories. Furthermore, incorporating $\mathcal{L}_D$ or adopting the voting technique enhances performance. The most optimal performance is achieved when all three components are used simultaneously, surpassing the second-place configuration (Table 1 row (C)) by nearly 8% in average performance.

## 6 CONCLUSION

This research uncovers the distinct and complementary strengths of the temporal and frequency domains in the context of time-series Unsupervised Domain Adaptation (UDA). Our initial experiments show that the temporal domain captures a wider range of discriminative features, while the

frequency domain focuses on domain-agnostic features that improve transferability between the source and target domains. Building on these findings, we introduce TidalFlow—an innovative SFUDA framework that effectively combines frequency embeddings and uses simple hyperparameter adjustments to adapt to new domains without relying on traditional alignment methods.

TidalFlow demonstrates significant performance improvements, achieving nearly a 10% gain across five benchmark datasets, highlighting its practical utility and robustness in real-world applications. By moving beyond conventional alignment-focused approaches, this work shifts the focus toward extracting class-specific features that remain consistent across domains. The methodologies and insights presented in this study represent a paradigm shift in time-series SFUDA, offering a more flexible and resilient framework that is better equipped to handle diverse and challenging domain adaptation scenarios.

**Limitation and future work.** Addressing issues related to class imbalances would be urgent for future research. Additionally, mitigating the frequency leakage problem, which can arise due to the integration of information from both time and frequency domains, is essential for further enhancing the model's performance. These endeavors will not only bolster TidalFlow's capabilities but also contribute valuable insights to the broader landscape of time-series SFUDA.

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

# A ALGORITHMS

An overview of TidalFlow is in Alg. 1. Moreover, we enhance the nearest neighbor algorithm of VQ-VAE to make it suitable for our UDA task. We utilize nearest neighbor function $\rho$ (Alg. 2) in both the training and adaptation phases, while voting function $V$ (Alg. 3) is applied during the inference stage. Unlike Alg. 1, we want to illustrate a more comprehensive explanation of implementation details, so both of these algorithms are implemented following the PyTorch style.

---

**Algorithm 1** Overview of TidalFlow

---

1: **Input:** data $x_i$; label $y_i^s$; Dual-stream encoder $E$; decoder $U$; classifier $C$; frequency block $B_S$; temporal block $B_T$; time step $T$; input channel $M$; nearest neighbor function $\rho$ (Alg. 2); voting function $V$ (Alg. 3)

---

2: Extract $z_e \leftarrow E(x_i)$

---

3: **First: Training Phase**
4:  Get $e_{h,j}, e_{p,q} \leftarrow \rho(z_e, [B_S; B_T], y_i^s)$
5:  $x_i' \leftarrow U(e_{h,j})$
6:  Compute objective functions $\mathcal{L}_{CE}$, $\mathcal{L}_A$ and $\mathcal{L}_D$
7:  Update $E, B_S, B_T$ and $C$ with
$$\nabla(\mathcal{L}_{CE} + \mathcal{L}_A + \mathcal{L}_D)$$

---

8: **Second: Adaptation Phase**
9:  Get $e_{p,q} \leftarrow \rho(z_e, [B_S; B_T])$
10:  $x_i' \leftarrow U(e_{p,q})$
11:  Compute objective functions $\mathcal{L}_{MSE}$ and $\mathcal{L}_A$
12:  Update $E, B_S, B_T$ and $U$ with $\nabla(\mathcal{L}_{MSE} + \mathcal{L}_A)$

---

13: **Third: Inference Phase**
14:  Get $e_{p',q'} \leftarrow V(z_e)$
15: **Output**: $p'$

---

864
865
866
867
868
869
870
871
872
873
874
875

---

**Algorithm 2** Finding Nearest Neighbor Function $\rho$

---

1: **Input:** Query $Q$, Target $T$, Labels $label$
2: **Initialization:**
3: $index\_list \leftarrow []$
4: $k \leftarrow$ Total embeddings for each category
5: $h \leftarrow$ Total classification categories
6: $Q \leftarrow Q.\text{unsqueeze}(1).\text{repeat}(1, k, 1)$
7: **for** $i = 1$ **to** $Q.\text{size}(0)$ **do**
8:     $T \leftarrow T[label[i] \times k : (label[i] + 1) \times k].\text{unsqueeze}(0)$
9:     $tmp\_index \leftarrow (Q[i] - T).\text{pow}(2).\text{sum}(2).\text{sqrt}().\text{min}(1)[1][0]$
10:     $index \leftarrow \text{int}(tmp\_index) + label[i] \times k$
11:     $index\_list.\text{append}(index)$
12: **end for**
13: $index\_tensor \leftarrow \text{torch.tensor}(index\_list)$
14: $e_{h,j} \leftarrow T[index\_tensor]$
15: **if** During Training Phase **then**
16:     {*Find the nearest neighbor from other categories.*}
17:     $index\_list \leftarrow []$
18:     $Q \leftarrow Q.\text{unsqueeze}(1).\text{repeat}(1, k \times (h - 1), 1)$
19:     **for** $i = 1$ **to** $Q.\text{size}(0)$ **do**
20:         $map\_original\_list \leftarrow \text{list}(\text{range}(k \times h))$
21:         del $map\_original\_list[label[i] \times k : (label[i] + 1) \times k]$
22:         $start\_index \leftarrow label[i] \times k$
23:         $end\_index \leftarrow (label[i] + 1) \times k$
24:         $target\_2 \leftarrow \text{torch.cat}((target[: start\_index], target[end\_index :]), \text{dim} = 0)$
25:         $T \leftarrow target\_2.\text{unsqueeze}(0)$
26:         $tmp\_index \leftarrow (Q[i] - T).\text{pow}(2).\text{sum}(2).\text{sqrt}().\text{min}(1)[1][0]$
27:         $index \leftarrow map\_original\_list[tmp\_index]$
28:         $index\_list.\text{append}(index)$
29:     **end for**
30:     $index\_tensor \leftarrow \text{torch.tensor}(index\_list)$
31:     $e_{p,q} \leftarrow T[index\_tensor]$
32:     **Output:** $e_{h,j}, e_{p,q}$
33: **else**
34:     **Output:** $e_{h,j}$
35: **end if**

---

908
909
910
911
912
913
914
915
916
917

---

**Algorithm 3** Voting Mechanism from Hierarchical Embedding Table

---

1: **Input:** Query $Q$.
2: **Initialization:**
3: $index\_list \leftarrow []$
4: $k \leftarrow$ Total embeddings for each category
5: $h \leftarrow$ Total classification categories
6: $Q \leftarrow Q.\text{unsqueeze}(1).\text{repeat}(1, \text{HET.size}(0), 1)$
7: $T \leftarrow \text{HET.unsqueeze}(0).\text{repeat}(Q.\text{size}(0), 1, 1)$
8: $indexes \leftarrow (Q - T).\text{pow}(2).\text{sum}(2).\text{sqrt}().\text{argsort}(\text{dim} = 1)[:, : 5]$
9: **for** $j$ **in** $indexes$ **do**
10:     $index \leftarrow j // hk$
11:     $counter \leftarrow \text{Counter}(index.\text{tolist}())$
12:     $most\_common\_index \leftarrow counter.\text{most\_common}(1)[0][0]$
13:     $index\_list.\text{append}(\text{int}(most\_common\_index))$
14: **end for**
15: $index\_tensor \leftarrow \text{torch.tensor}(index\_list)$
16: **Output:** $T[index\_tensor]$

---

## B  DATASET DETAILS FOR UDA BENCHMARK

We assess the performance of TidalFlow on five distinct UDA benchmark datasets, each characterized by its unique features. The datasets considered include:

1. HAR Anguita et al. (2013): This dataset incorporates measurements from a 3-axis accelerometer, 3-axis gyroscope, and 3-axis body acceleration. Data is collected from 30 participants at a sampling rate of 50 Hz and uses non-overlapping segments of 128-time steps to predict activity labels. The objective is to classify time series into six activities: walking, walking upstairs, walking downstairs, sitting, standing, and lying down.

2. HHAR Stisen et al. (2015): Comprising 3-axis accelerometer measurements from 9 participants at a frequency of 50 Hz, this dataset employs non-overlapping segments of 128-time steps for classification. Activity labels include biking, sitting, standing, walking, walking upstairs, and walking downstairs.

3. WISDM Kwapisz et al. (2011): Featuring 3-axis accelerometer measurements from 36 participants at a frequency of 20 Hz, similar to the HAR dataset, we use non-overlapping segments of 128-time steps for classification. The dataset includes six activity labels: walking, jogging, sitting, standing, walking upstairs, and walking downstairs.

4. Sleep-EDF Goldberger et al. (2000): This task involves classifying electroencephalography (EEG) signals into five stages (Wake, N1, N2, N3, REM). Comprising EEG readings from 20 healthy subjects, we select a single channel (Fpz-Cz) as Ragab et al. (2023a).

5. MFD Lessmeier et al. (2016): Collected by Paderborn University to identify incipient faults using vibration signals, this dataset consists of data collected under four different operating conditions. Each condition is treated as a separate domain, and we use five cross-condition scenarios to evaluate domain adaptation performance. Each sample in the dataset comprises a single univariate channel with 5120 data points.

The summary of the datasets is in Table 2. These datasets span diverse applications and challenges, enabling a comprehensive evaluation of TidalFlow's effectiveness and robustness across various domains.

## C  UDA ON BENCHMARK DATASETS

We engage in activity prediction through an Unsupervised Domain Adaptation approach, utilizing benchmark datasets such as HAR, HHAR, and WISDM. Additionally, we delve into specific tasks within the medical and mechanical engineering domains, focusing on the Sleep-EDF and MFD datasets, respectively.

Table 2: Summary of datasets. Ragab et al. (2023a)

| Dataset | #Subjects/Domains | #Class | #Channels | Length | #Train | #Test |
|---|---|---|---|---|---|---|
| HAR | 30 | 6 | 9 | 128 | 2300 | 990 |
| HHAR | 9 | 6 | 3 | 128 | 12716 | 5218 |
| WISDM | 30 | 6 | 3 | 128 | 1350 | 720 |
| Sleep-EDF | 20 | 5 | 1 | 3000 | 14280 | 6310 |
| MFD | 4 | 3 | 1 | 5120 | 7312 | 3604 |

For each dataset, we present prediction results for 10 randomly selected source $\mapsto$ target pairs. To ensure robustness, we conduct the experiments with 5 random initializations and report the mean and standard deviation values. The results are organized into tables:

- Table 3: Mean accuracy and average Macro-F1 on the target domains for the HAR dataset.
- Table 4: Mean accuracy and average Macro-F1 on the target domains for the HHAR dataset.
- Table 5: Mean accuracy and average Macro-F1 on the target domains for the WISDM dataset.
- Table 6: Mean accuracy and average Macro-F1 on the target domains for the Sleep-EDF dataset.
- Table 7: Mean accuracy and average Macro-F1 on the target domains for the MFD dataset.

Table 3: Prediction accuracy for HAR Dataset between various subjects. Shown: mean accuracy and macro F1 over 5 random initializations.

| METHOD | $2 \mapsto 9$ | $1 \mapsto 14$ | $1 \mapsto 10$ | $4 \mapsto 9$ | $21 \mapsto 29$ | $25 \mapsto 28$ | $30 \mapsto 2$ | $4 \mapsto 3$ | $2 \mapsto 11$ | $9 \mapsto 18$ |
|---|---|---|---|---|---|---|---|---|---|---|
| | | | | | MEAN ACCURACY (%) | | | | | |
| AVG | 59.58 | 73.26 | 53.64 | 61.62 | 73.17 | 82.92 | 59.62 | 88.54 | 85.94 | 60.75 |
| STD OF AVG | 11.73 | 11.30 | 11.99 | 11.38 | 16.07 | 5.42 | 16.58 | 11.99 | 11.15 | 14.69 |
| W/O UDA | 48.28 | 81.44 | 52.81 | 68.97 | 50.96 | 84.35 | 54.95 | 66.02 | 77.89 | 30.91 |
| DEEPCORAL | 50.63 | 75.00 | 57.50 | 58.44 | 76.25 | 82.91 | 46.87 | 93.12 | 90.63 | 46.88 |
| CDAN | 66.88 | _88.95_ | 56.87 | 63.13 | **89.58** | 85.21 | 54.37 | 97.29 | 85.42 | 58.86 |
| DIRT-T | 69.68 | 60.62 | **62.81** | 52.81 | 85.62 | 74.37 | 55.00 | 84.58 | 80.21 | 59.03 |
| HOMM | 35.00 | 58.96 | 23.75 | 37.81 | 39.37 | 73.75 | 41.88 | 72.71 | 65.47 | 41.27 |
| CODATS | 59.06 | 79.58 | 54.69 | 67.50 | 81.87 | _88.75_ | 71.56 | 88.12 | 68.23 | 63.89 |
| ADVSKM | 51.25 | 78.54 | 57.19 | 59.06 | 76.67 | 84.37 | 47.18 | 91.04 | _98.96_ | 74.65 |
| CLUDA | 65.91 | 57.14 | 42.22 | 50.00 | 61.54 | 74.14 | 52.17 | _98.08_ | 81.77 | 67.71 |
| RAINCOAT | _70.31_ | 63.54 | _62.50_ | _73.13_ | 84.16 | _88.75_ | **87.50** | 96.46 | **100.0** | _75.69_ |
| **OURS** | **73.12** | **90.01** | 61.87 | **80.08** | _87.23_ | **88.79** | _86.94_ | **100.0** | **100.0** | **76.17** |
| | | | | | MEAN MACRO F1 | | | | | |
| AVG | 0.538 | 0.709 | 0.539 | 0.601 | 0.686 | 0.822 | 0.593 | 0.877 | 0.833 | 0.580 |
| STD OF AVG | 0.119 | 0.120 | 0.120 | 0.113 | 0.207 | 0.068 | 0.133 | 0.125 | 0.140 | 0.150 |
| W/O UDA | 0.374 | 0.802 | 0.524 | 0.685 | 0.351 | 0.840 | 0.500 | 0.569 | 0.714 | 0.190 |
| DEEPCORAL | 0.440 | 0.733 | 0.590 | 0.554 | 0.714 | 0.832 | 0.492 | 0.927 | 0.910 | 0.440 |
| CDAN | 0.621 | _0.879_ | 0.591 | 0.642 | **0.900** | 0.846 | 0.523 | 0.969 | 0.850 | 0.610 |
| DIRT-T | _0.675_ | 0.501 | _0.645_ | 0.458 | 0.861 | 0.706 | 0.491 | 0.811 | 0.810 | 0.580 |
| HOMM | 0.313 | 0.550 | 0.224 | 0.318 | 0.296 | 0.730 | 0.453 | 0.677 | 0.573 | 0.366 |
| CODATS | 0.538 | 0.789 | 0.538 | 0.685 | 0.797 | **0.899** | 0.721 | 0.866 | 0.660 | 0.600 |
| ADVSKM | 0.452 | 0.767 | 0.583 | 0.549 | 0.737 | 0.846 | 0.519 | 0.893 | _0.990_ | _0.730_ |
| CLUDA | 0.664 | 0.557 | 0.389 | 0.511 | 0.570 | 0.756 | 0.481 | _0.980_ | 0.810 | 0.670 |
| RAINCOAT | 0.645 | 0.614 | 0.626 | _0.724_ | 0.831 | _0.899_ | **0.864** | 0.963 | **1.000** | **0.760** |
| **OURS** | **0.727** | **0.888** | **0.649** | **0.778** | _0.894_ | **0.905** | _0.848_ | **1.000** | **1.000** | 0.728 |

Table 4: Prediction accuracy for HHAR Dataset between various subjects. Shown: mean accuracy and macro F1 over 5 random initializations.

| | MEAN ACCURACY (%) | | | | | | | | | |
|---|---|---|---|---|---|---|---|---|---|---|
| METHOD | $7 \mapsto 6$ | $1 \mapsto 3$ | $0 \mapsto 2$ | $2 \mapsto 3$ | $2 \mapsto 6$ | $7 \mapsto 2$ | $4 \mapsto 0$ | $5 \mapsto 0$ | $7 \mapsto 0$ | $4 \mapsto 2$ |
| AVG | 88.96 | 93.93 | 78.17 | 56.28 | 44.35 | 38.85 | 32.81 | 33.31 | 32.75 | 26.37 |
| STD OF AVG | 6.92 | 5.21 | 7.15 | 7.33 | 8.75 | 5.30 | 7.49 | 6.85 | 7.63 | 6.33 |
| W/O UDA | 78.04 | **98.51** | 64.51 | 50.32 | 45.11 | 32.37 | 32.81 | 30.42 | 33.92 | 19.16 |
| DEEPCORAL | 79.08 | 88.24 | 84.23 | 54.32 | 45.28 | 34.45 | 28.13 | 42.04 | 38.62 | 23.74 |
| CDAN | 96.04 | 93.01 | 76.19 | 60.27 | 31.88 | 37.05 | 29.09 | 22.84 | 25.09 | 27.16 |
| DIRT-T | 93.79 | 95.09 | 77.83 | **66.22** | 50.69 | 38.10 | 32.22 | 24.70 | 27.81 | 26.41 |
| HOMM | 84.63 | 88.91 | 68.38 | 45.83 | 44.03 | 35.94 | 32.37 | 34.60 | 29.60 | 23.21 |
| CODATS | 88.95 | 95.16 | 79.61 | 61.09 | 35.90 | 38.54 | 21.80 | 33.85 | 32.41 | **36.31** |
| ADVSKM | 83.71 | 82.07 | 78.94 | 43.45 | 36.67 | 39.95 | 33.49 | 34.60 | 24.91 | 19.05 |
| CLUDA | 92.43 | 96.51 | 79.84 | 59.83 | 56.18 | 37.80 | 38.84 | 34.93 | 44.59 | 35.29 |
| RAINCOAT | 89.90 | 95.65 | **87.82** | 60.04 | 40.21 | 43.32 | **46.46** | 30.36 | 27.90 | 24.33 |
| **OURS** | **97.04** | 96.91 | 87.54 | 65.78 | **57.01** | **51.46** | 46.28 | **42.38** | **44.97** | 35.31 |
| | MEAN MACRO F1 | | | | | | | | | |
| AVG | 0.882 | 0.930 | 0.738 | 0.514 | 0.400 | 0.374 | 0.327 | 0.293 | 0.343 | 0.247 |
| STD OF AVG | 0.069 | 0.056 | 0.091 | 0.081 | 0.068 | 0.061 | 0.083 | 0.067 | 0.064 | 0.075 |
| W/O UDA | 0.783 | **0.985** | 0.600 | 0.410 | 0.359 | 0.310 | 0.290 | 0.220 | 0.337 | 0.135 |
| DEEPCORAL | 0.761 | 0.874 | 0.860 | 0.498 | 0.419 | 0.320 | 0.260 | 0.380 | 0.409 | 0.230 |
| CDAN | 0.961 | 0.930 | 0.700 | 0.563 | 0.325 | 0.320 | 0.270 | 0.202 | 0.265 | 0.257 |
| DIRT-T | 0.936 | 0.950 | 0.760 | **0.628** | 0.441 | 0.340 | 0.300 | 0.207 | 0.303 | 0.283 |
| HOMM | 0.836 | 0.881 | 0.625 | 0.408 | 0.398 | 0.377 | 0.318 | 0.306 | 0.315 | 0.192 |
| CODATS | 0.883 | 0.951 | 0.730 | 0.580 | 0.366 | 0.360 | 0.200 | 0.328 | 0.315 | **0.356** |
| ADVSKM | 0.821 | 0.791 | 0.720 | 0.388 | 0.333 | 0.410 | 0.330 | 0.279 | 0.270 | 0.157 |
| CLUDA | 0.928 | 0.965 | 0.820 | 0.544 | 0.506 | 0.360 | 0.400 | 0.305 | 0.426 | 0.345 |
| RAINCOAT | 0.903 | 0.955 | **0.870** | 0.553 | 0.397 | 0.440 | 0.450 | 0.288 | 0.331 | 0.235 |
| **OURS** | **0.987** | 0.967 | 0.814 | 0.611 | **0.544** | **0.518** | **0.453** | **0.409** | **0.444** | 0.381 |

Table 5: Prediction accuracy for WISDM Dataset between various subjects. Shown: mean accuracy and macro F1 over 5 random initializations.

| METHOD | 4 ↦ 5 | 11 ↦ 16 | 12 ↦ 23 | 18 ↦ 23 | 26 ↦ 29 | 28 ↦ 27 | 4 ↦ 11 | 28 ↦ 21 | 12 ↦ 26 | 17 ↦ 26 |
|---|---|---|---|---|---|---|---|---|---|---|
| | MEAN ACCURACY (%) | | | | | | | | | |
| AVG | 64.93 | 17.12 | 50.47 | 50.07 | 28.67 | 60.00 | 42.57 | 47.52 | 52.60 | 59.65 |
| STD OF AVG | 11.59 | 10.01 | 13.26 | 16.15 | 14.17 | 23.59 | 11.50 | 20.36 | 8.73 | 9.65 |
| W/O UDA | 42.03 | 13.73 | 45.00 | 58.33 | **50.00** | 8.00 | 32.89 | 59.62 | 54.88 | 43.90 |
| DEEPCORAL | 76.81 | 15.69 | 39.17 | 61.67 | 21.67 | 68.00 | 27.63 | 28.85 | 48.17 | 65.24 |
| CDAN | 60.87 | 17.65 | 61.67 | 23.33 | 15.00 | 76.00 | 44.74 | 61.54 | 48.78 | 65.85 |
| DIRT-T | 73.91 | 6.86 | 63.33 | 56.67 | 39.17 | 46.00 | 42.11 | 41.35 | 53.66 | 63.41 |
| HOMM | 57.97 | 3.92 | 32.50 | 45.83 | 39.17 | 52.00 | 32.24 | 31.73 | 40.85 | 43.90 |
| CODATS | 56.52 | 30.39 | 52.50 | 60.83 | 27.50 | 66.00 | 54.61 | 31.73 | 64.02 | **70.12** |
| ADVSKM | 61.59 | 23.53 | 29.17 | 25.00 | 36.67 | 78.00 | 24.34 | 17.31 | 35.98 | 56.71 |
| CLUDA | 62.86 | 15.38 | 54.84 | 48.39 | 6.67 | 36.00 | 47.37 | 34.62 | 48.78 | 51.22 |
| RAINCOAT | 65.22 | 19.61 | 63.33 | 63.33 | 21.67 | 84.00 | 43.42 | 84.62 | 57.32 | 64.63 |
| **OURS** | **87.96** | **42.32** | **66.77** | **69.69** | 49.75 | **85.21** | **72.58** | **84.64** | **64.04** | 65.77 |
| | MEAN MACRO F1 | | | | | | | | | |
| AVG | 0.515 | 0.170 | 0.298 | 0.281 | 0.191 | 0.403 | 0.328 | 0.389 | 0.257 | 0.391 |
| STD OF AVG | 0.178 | 0.094 | 0.137 | 0.114 | 0.067 | 0.183 | 0.136 | 0.204 | 0.046 | 0.149 |
| W/O UDA | 0.099 | 0.083 | 0.176 | 0.226 | 0.133 | 0.033 | 0.329 | 0.388 | 0.223 | 0.160 |
| DEEPCORAL | 0.704 | 0.166 | 0.176 | 0.308 | 0.136 | 0.519 | 0.300 | 0.225 | 0.234 | 0.456 |
| CDAN | 0.366 | 0.277 | 0.340 | 0.156 | 0.218 | 0.337 | 0.383 | 0.541 | 0.257 | 0.422 |
| DIRT-T | 0.492 | 0.096 | 0.382 | 0.274 | 0.255 | 0.496 | 0.276 | 0.346 | 0.255 | 0.417 |
| HOMM | 0.501 | 0.020 | 0.201 | 0.268 | 0.268 | 0.421 | 0.229 | 0.245 | 0.237 | 0.281 |
| CODATS | 0.496 | **0.283** | 0.384 | 0.508 | 0.151 | 0.291 | 0.414 | 0.266 | 0.310 | 0.502 |
| ADVSKM | 0.548 | 0.271 | 0.191 | 0.160 | 0.269 | 0.458 | 0.204 | 0.154 | 0.221 | 0.438 |
| CLUDA | 0.611 | 0.126 | 0.359 | 0.275 | 0.111 | 0.370 | 0.262 | 0.321 | 0.236 | 0.233 |
| RAINCOAT | 0.461 | 0.265 | **0.519** | 0.283 | 0.162 | **0.713** | 0.333 | 0.691 | 0.267 | 0.398 |
| **OURS** | **0.819** | 0.254 | 0.517 | **0.548** | **0.311** | 0.588 | **0.705** | **0.730** | **0.369** | **0.731** |

Table 6: Prediction accuracy for Sleep-EDF Dataset between various subjects. Shown: mean accuracy and macro F1 over 5 random initializations.

| METHOD | $1 \mapsto 8$ | $6 \mapsto 10$ | $8 \mapsto 0$ | $2 \mapsto 1$ | $15 \mapsto 4$ | $8 \mapsto 1$ | $4 \mapsto 19$ | $8 \mapsto 5$ | $18 \mapsto 6$ | $13 \mapsto 7$ |
|---|---|---|---|---|---|---|---|---|---|---|
| | | | | | MEAN ACCURACY (%) | | | | | |
| AVG | 57.07 | 71.17 | 67.77 | 75.71 | 69.42 | 62.65 | 72.76 | 54.02 | 72.34 | 65.07 |
| STD OF AVG | 8.76 | 8.14 | 8.49 | 6.35 | 4.12 | 7.25 | 8.42 | 12.00 | 9.69 | 8.03 |
| W/O UDA | 52.05 | 75.11 | 68.53 | 78.75 | 68.54 | 61.43 | 77.58 | 51.39 | 76.14 | 68.44 |
| DEEPCORAL | 61.82 | 71.09 | 66.41 | 78.07 | 69.90 | 62.66 | 72.74 | 43.62 | 76.17 | 68.85 |
| CDAN | 45.62 | 75.31 | 75.13 | 73.23 | 70.78 | 60.16 | 68.97 | 65.89 | 75.78 | 65.62 |
| DIRT-T | 49.06 | 77.97 | 84.83 | 77.92 | 68.75 | 69.84 | 80.56 | 70.25 | 72.72 | 61.77 |
| HOMM | 62.29 | 71.61 | 64.58 | 65.05 | 73.70 | 58.70 | 67.62 | 36.91 | 76.43 | 66.46 |
| CODATS | 62.55 | 67.29 | 62.63 | 79.74 | 72.71 | 60.57 | 82.34 | 55.01 | 68.82 | 75.00 |
| ADVSKM | 67.34 | 71.20 | 59.31 | 79.53 | 69.32 | 60.26 | 70.62 | 38.35 | 74.09 | 66.04 |
| CLUDA | 46.81 | 53.64 | 51.01 | 60.47 | 57.65 | 45.64 | 48.58 | 43.40 | 47.31 | 47.93 |
| RAINCOAT | 59.74 | 77.08 | 72.98 | 78.33 | 69.90 | 66.30 | 71.83 | 64.78 | 76.17 | 65.78 |
| **OURS** | **75.81** | **78.66** | **78.97** | **80.13** | **73.65** | **76.92** | **82.51** | **73.84** | **80.52** | **77.98** |
| | | | | | MEAN MACRO F1 | | | | | |
| AVG | 0.498 | 0.567 | 0.596 | 0.664 | 0.609 | 0.546 | 0.564 | 0.517 | 0.618 | 0.570 |
| STD OF AVG | 0.110 | 0.137 | 0.094 | 0.135 | 0.080 | 0.108 | 0.156 | 0.118 | 0.137 | 0.093 |
| W/O UDA | 0.409 | 0.694 | 0.632 | 0.677 | 0.564 | 0.560 | 0.619 | 0.559 | 0.651 | 0.576 |
| DEEPCORAL | 0.556 | 0.574 | 0.582 | 0.728 | 0.640 | 0.565 | 0.618 | 0.464 | 0.670 | 0.611 |
| CDAN | 0.400 | 0.590 | 0.636 | 0.687 | 0.596 | 0.495 | 0.529 | 0.573 | 0.664 | 0.572 |
| DIRT-T | 0.445 | 0.596 | 0.714 | 0.710 | 0.583 | 0.563 | 0.671 | 0.590 | 0.618 | 0.523 |
| HOMM | 0.548 | 0.582 | 0.572 | 0.662 | **0.691** | 0.540 | 0.551 | 0.402 | 0.643 | 0.591 |
| CODATS | 0.555 | 0.534 | 0.522 | 0.696 | 0.668 | 0.497 | 0.719 | 0.489 | 0.627 | 0.630 |
| ADVSKM | 0.599 | 0.545 | 0.519 | **0.740** | 0.656 | 0.562 | 0.587 | 0.401 | 0.650 | 0.607 |
| CLUDA | 0.310 | 0.179 | 0.364 | 0.338 | 0.409 | 0.305 | 0.233 | 0.305 | 0.284 | 0.365 |
| RAINCOAT | 0.528 | 0.641 | 0.601 | 0.724 | 0.578 | 0.572 | 0.536 | 0.540 | 0.675 | 0.527 |
| **OURS** | **0.715** | **0.749** | **0.702** | 0.702 | 0.645 | **0.790** | **0.751** | **0.757** | **0.720** | **0.665** |

Table 7: Prediction accuracy for MFD Dataset between various subjects. Shown: mean accuracy and macro F1 over 5 random initializations.

| METHOD | $0 \mapsto 1$ | $0 \mapsto 3$ | $1 \mapsto 2$ | $1 \mapsto 0$ | $3 \mapsto 0$ | $2 \mapsto 0$ | $3 \mapsto 2$ | $0 \mapsto 2$ | $2 \mapsto 1$ | $1 \mapsto 3$ |
|---|---|---|---|---|---|---|---|---|---|---|
| | | | | | MEAN ACCURACY (%) | | | | | |
| AVG | 58.27 | 65.47 | 70.47 | 54.70 | 56.08 | 51.09 | 69.53 | 61.15 | 79.43 | 87.18 |
| STD OF AVG | 9.92 | 9.10 | 10.24 | 14.13 | 14.71 | 13.39 | 11.82 | 4.27 | 15.30 | 14.54 |
| W/O UDA | 41.73 | 51.39 | 67.04 | 42.06 | 39.84 | 28.97 | 79.69 | 61.71 | 88.46 | 98.45 |
| DEEPCORAL | 66.15 | 69.79 | 64.21 | 41.67 | 48.33 | 41.67 | 61.53 | 65.89 | 89.14 | 81.32 |
| CDAN | 47.36 | 68.79 | 76.00 | 46.61 | 50.04 | 49.33 | 70.24 | 62.69 | 90.62 | 99.44 |
| DIRT-T | 58.37 | 65.62 | 72.19 | **81.10** | 73.40 | **70.65** | 74.63 | 64.84 | 70.83 | 98.85 |
| HOMM | 65.59 | 68.34 | 65.29 | 42.56 | 47.84 | 36.64 | 62.35 | 59.90 | 82.66 | 81.81 |
| CODATS | 60.66 | 62.72 | **86.16** | 41.74 | 45.59 | 42.58 | 79.97 | 54.91 | 81.03 | **100.0** |
| ADVSKM | 64.73 | 71.80 | 65.10 | 40.85 | 48.25 | 45.05 | 61.87 | 64.14 | 86.24 | 82.63 |
| CLUDA | 48.34 | 48.56 | 48.12 | 41.69 | 42.57 | 47.67 | 49.45 | 54.77 | 46.56 | 44.79 |
| RAINCOAT | 63.02 | 67.49 | 76.45 | 61.53 | 68.45 | 65.40 | 81.55 | 58.82 | **92.30** | 97.14 |
| **OURS** | **73.96** | **84.28** | 83.51 | 78.77 | **84.98** | 67.24 | **87.22** | 67.33 | 91.71 | 99.84 |
| | | | | | MEAN MACRO F1 | | | | | |
| AVG | 0.480 | 0.565 | 0.736 | 0.541 | 0.581 | 0.537 | 0.734 | 0.548 | 0.828 | 0.896 |
| STD OF AVG | 0.083 | 0.108 | 0.164 | 0.189 | 0.158 | 0.125 | 0.169 | 0.108 | 0.205 | 0.258 |
| W/O UDA | 0.400 | 0.520 | 0.758 | 0.575 | 0.558 | 0.479 | 0.851 | 0.674 | 0.915 | 0.989 |
| DEEPCORAL | 0.496 | 0.551 | 0.688 | 0.477 | 0.503 | 0.473 | 0.667 | 0.607 | 0.919 | 0.856 |
| CDAN | 0.318 | 0.523 | 0.800 | 0.343 | 0.428 | 0.452 | 0.743 | 0.525 | 0.925 | 0.996 |
| DIRT-T | 0.492 | 0.634 | 0.788 | **0.830** | 0.756 | 0.742 | 0.789 | **0.733** | 0.777 | 0.992 |
| HOMM | 0.460 | 0.490 | 0.700 | 0.480 | 0.501 | 0.424 | 0.665 | 0.442 | 0.866 | 0.859 |
| CODATS | 0.557 | **0.689** | **0.871** | 0.451 | 0.532 | 0.499 | 0.826 | 0.393 | 0.843 | **1.000** |
| ADVSKM | 0.450 | 0.633 | 0.685 | 0.473 | 0.504 | 0.501 | 0.674 | 0.560 | 0.896 | 0.866 |
| CLUDA | 0.408 | 0.339 | 0.333 | 0.252 | 0.295 | 0.323 | 0.345 | 0.383 | 0.325 | 0.311 |
| RAINCOAT | **0.610** | 0.655 | 0.806 | 0.692 | 0.737 | 0.719 | 0.850 | 0.581 | **0.941** | 0.979 |
| **OURS** | 0.580 | 0.623 | **0.871** | 0.826 | **0.885** | **0.751** | **0.902** | 0.577 | 0.925 | 0.993 |

# D  IMPLEMENTATION DETAILS FOR HYPERPARAMETERS

## D.1  LEARNING RATE

Table 8: Leaning rates of different components in TidalFlow.

| Component | Training Phase | Adaptation Phase |
|---|---|---|
| Encoder | 1e-4 | 2e-6 |
| HET - temporal block | 2e-4 | 2e-6 |
| HET - frequency block | 2e-4 | 1e-8 |
| Classifier | 1e-2 | - |
| Decoder | - | 2e-4 |

## D.2  TRAINING BATCH SIZE

Table 9: Batch sizes of different datasets in TidalFlow.

| Dataset | Training Phase | Adaptation Phase |
|---|---|---|
| HAR | 32 | 32 |
| HHAR | 32 | 32 |
| WISDM | 32 | 32 |
| Sleep-EDF | 32 | 32 |
| MFD | 32 | 32 |

## D.3  PARAMETER $K$

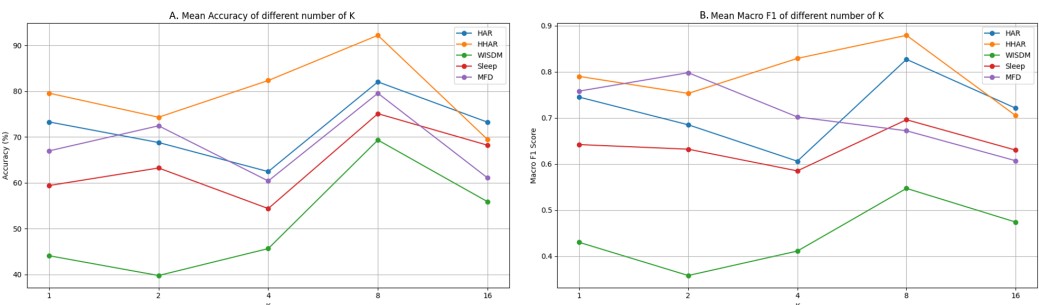

Figure 6: TidalFlow's performance in different $K$. We observed that for the majority of datasets, setting $K$ to 8 yielded better performance, excluding MFD dataset in mean macro F1 score.

## D.4  PARAMETER $\gamma$

Table 10: $\gamma$ in different datasets.

| HAR | HHAR | WISDM | Sleep-EDF | MFD |
|---|---|---|---|---|
| 1.2 | 1.2 | 1 | 1 | 1.5 |

# E    COMPUTATION ANALYSIS

Two main factors affect TidalFlow's performance: (1) the size of the hierarchical embedding table and (2) the number of classification categories. The following will elaborate on these two aspects:

## E.1    SIZE OF THE HIERARCHICAL EMBEDDING TABLE

During the training phase, as the source domain has labels, we only need to calculate $K$ nearest neighbors for each category, where $K$ represents the number of embeddings per category (Fig. 6). We determine the appropriate value of $K$ through experimentation, considering both Mean accuracy and macro F1 score. We found that for the majority of datasets, setting $K$ to 8 yielded better performance, excluding MFD dataset in mean macro F1 score. Accordingly, we speculate that other parameters of the model contribute to its superior performance at $K$=8.

## E.2    NUMBER OF CLASSIFICATION CATEGORIES

During the adaptation phase, as the target domain lacks labels, we must compute all embeddings in the embedding table to obtain the closest embeddings. At this point, the time required by the model is directly influenced by the number of categories, leading to a significant impact.

Our study utilized an A100 GPU 40GB, with an average total training time of 0.5 GPU hours across the five datasets. Table 11 is the relevant parameter table for the 5 datasets:

Table 11: Epochs of training and adaptation phases in different datasets.

| DATASET | TRAINING EPOCH | ADAPTATION EPOCH |
|---|---|---|
| HAR | 70 | 50 |
| HHAR | 80 | 70 |
| WISDM | 150 | 50 |
| SLEEP-EDF | 200 | 100 |
| MFD | 150 | 100 |

# F    BROADER IMPACTS

**Potential positive societal impacts.** We may apply TidalFlow in smart elderly care facilities. Given the significant differences in behavior between the elderly population and middle-aged adults, such as frequent nocturnal bathroom visits, slower mobility, and increased susceptibility to falls, leveraging the human activity recognition datasets (i.e, HAR, HHAR, WISDM, DSADS) as the source domain and adapting it to the elderly population for downstream tasks could be a crucial research direction and technological advancement in the future.

**Potential negative societal impacts.** As our task involves domain adaptation, there are no noteworthy negative social impacts to consider.

