# OpenReview forum: "Rethinking the Roles of Time and Frequency Domains Before Tackling Time Series UDA"
_ICLR.cc/2025/Conference — Submitted to ICLR 2025_

### Official Review · Reviewer_nRL7 · 2024-10-21

**Soundness:** 3
**Presentation:** 3
**Contribution:** 3
**Rating:** 6
**Confidence:** 4

**Summary:**

The paper addresses the challenge of time-series unsupervised domain adaptation (UDA) by exploring the roles of temporal and frequency domains, which have been relatively underexplored in this field. Through a series of experiments, the authors find that the temporal domain contains diverse, discriminative features but is sensitive to domain shifts, while the frequency domain offers more domain-invariant features that aid in transferability.

**Strengths:**

The paper presents a novel perspective on time-series unsupervised domain adaptation (UDA) by emphasizing the complementary strengths of temporal and frequency domains. While most existing UDA approaches primarily focus on aligning features between source and target domains, TidalFlow shifts the focus toward leveraging class-specific features from both domains without relying on traditional alignment strategies. This approach is original and opens up new avenues for UDA, moving beyond feature alignment to effective feature extraction and integration, which can be more robust to domain shifts.

The paper is clear and well-organized. The problem statement and motivation are explicitly articulated, helping readers understand why focusing on both temporal and frequency domains is essential. The descriptions of the TidalFlow framework, including its components like the dual-stream encoder, hierarchical embedding table (HET), and voting mechanism, are well-detailed. The visualizations and figures (e.g., PCA plots, performance charts) effectively support the explanations, making the results easy to interpret. The step-by-step presentation of the methodology and the logical flow between sections ensure that readers can follow the paper without confusion.

**Weaknesses:**

One of the main weaknesses of the paper is the absence of a thorough analysis of the computational cost associated with TidalFlow. Given the integration of both temporal and frequency domains, and the reliance on a dual-stream encoder and hierarchical embedding table (HET), there could be increased computational demands, especially when dealing with large-scale datasets or longer time series. For instance, it is not clear how the method scales when the number of categories \(H\) or the dimensionality of embeddings \(\Psi\) grows. The authors should consider providing a detailed assessment of time and memory usage, especially when comparing TidalFlow against other baseline methods. This will help practitioners understand the potential trade-offs between performance and computational requirements.

While the paper discusses the impact of different learning rates for temporal and frequency blocks, it does not comprehensively explore other hyperparameters. Key aspects such as the size of the embedding table, the number of nearest neighbors used in the voting mechanism, and the values of \(\alpha\) and \(\beta\) weights in the loss functions could significantly affect performance. Providing a more extensive sensitivity analysis on these hyperparameters would make it easier to understand the robustness of the model and help users in real-world applications to fine-tune the model effectively.

The effectiveness of certain components, such as the voting mechanism and the use of dissimilarity loss \(L_D\), is shown through experimental results, but there is limited explanation regarding why these components lead to performance improvements. For example, the voting mechanism is found to enhance classification performance, but the underlying reasons for its robustness are not thoroughly discussed. Similarly, it is unclear why the absence of the frequency block and \(L_D\) leads to such a significant drop in performance (as seen in the ablation study). Providing more theoretical insights or empirical explanations for these effects would strengthen the overall contribution of the paper and make the proposed model's design more justifiable.

By addressing these points, the paper can offer a more robust and practical understanding of TidalFlow, making it more appealing to both researchers and practitioners.

**Questions:**

see weakness

---

### Official Review · Reviewer_VLzi · 2024-10-23

**Soundness:** 2
**Presentation:** 3
**Contribution:** 3
**Rating:** 5
**Confidence:** 5

**Summary:**

This paper introduces TidalFlow, a novel framework for unsupervised domain adaptation (UDA) in time-series data. TidalFlow strategically integrates both temporal and frequency domain features to enhance UDA prediction performance. The authors demonstrate that the temporal domain provides richer, more discriminative features, while the frequency domain offers better domain-invariant properties, enhancing transferability across domains. Through comprehensive experiments on five benchmark datasets, TidalFlow shows nearly a 10% performance improvement over existing methods. The paper also provides a detailed analysis of the complementary strengths of the temporal and frequency domains, offering a new perspective on their roles in UDA.

**Strengths:**

1. This paper rethinks the roles of time and frequency domains by conducting thorough experiments, offering valuable insights into the advantages of integrating frequency information into the MTS-UDA task, and effectively demonstrating its impact and rationale.

2. The framework has been evaluated on five benchmark datasets, which provides sufficient validation.

**Weaknesses:**

1. In the related works section, it would be beneficial to provide a brief categorization or summary table that groups the related works by their primary application domain (e.g. video, speech, computer vision, time-series).

2. Typo in line 175: repeat "until".

3. Although the authors reassess the characteristics of temporal and frequency information in the 3.2 preliminary study, certain design elements in the main methodology are quite straightforward and not clearly explained. (please kindly refer to Quesiont (2))

4. For reproducibility, could you please include a table in the appendix listing key hyperparameters (e.g. learning rates, batch sizes, number of epochs) for all models, including baselines, to ensure fair comparison?

**Questions:**

(1) The Fourier transform is utilized, but it is unclear why this approach was chosen over alternatives like the wavelet transform. Could you briefly discuss the tradeoffs between Fourier transform and other time-frequency analysis methods like wavelet transform in the context of their specific task? This would encourage a more comprehensive justification of their methodological choices.

(2) The reasoning behind the framework's design is not explained clearly. What are the dimensions of the frequency input and the output from each module? How is the structure of G encoded? What is the rationale for the 'reconstruction' in the adaptation process?

(3) What does the training pipeline look like? Are the training and adaptation phases executed simultaneously, or does the adaptation phase occur only after the training phase is finished? If it's the latter, how is the completion of the adaptation phase determined?

(4) In the appendices, tables 3, 4, 5, 6, and 7, why are the average and standard deviation calculated across all models for each source-target pair? Could you explain the rationale for this aggregation or revise the tables to show per-model statistics, which would indeed facilitate easier comparison between models?

---

### Official Review · Reviewer_JdT6 · 2024-10-30

**Soundness:** 2
**Presentation:** 3
**Contribution:** 2
**Rating:** 5
**Confidence:** 5

**Summary:**

This paper proposes a new unsupervised domain adaptation framework (TidalFlow) for time series data. The authors revisit the roles of the time and frequency representations of time series data and suggest that the time domain offers higher discriminability, while the frequency domain provides better transferability.
In addition, the authors are interested in class-wise alignment between source and target domains besides the traditional global alignment.

**Strengths:**

- The paper is well written
- The experiments are extensive

**Weaknesses:**

1. First, I find that the paper is more suited to source-free UDA settings than traditional UDA. This is because source and target domains are not trained together. In contrast, the model is first pretrained on the source domain data, then adapted somehow to the target domain, which is similar to the source-free UDA settings.

2. I have a concern about the preliminary study, i.e., relying on the WISDM dataset. The reason is that this dataset is too small and its results are not very stable, and you can easily get 100% accuracy.
Also, using the accuracy instead of the F1-score for this highly imbalanced dataset is concerning.
Therefore, I find that the conclusions from this experiment unreliable.

3. In the problem formulation section, you never mentioned what is the objective. What do we do with the datasets you defined?
4. Most of the references are incomplete and missing the venue. This is common when you grab the citation from Arxiv as @misc. So please, get the references from the publishing venue source.

4. Please consider the following:
     - Fonts on figures, eg. Fig 1 are too small.
     - Typo: until until

**Questions:**

- I suggest that you reconsider the paper scenario to source-free UDA.
- I suggest that you rely, in the Preliminary experiment, on another more reliable dataset. Also, it would be better to use F1-score in imbalanced datasets.
- In the above study, you have one experiment adjusting the learning rate during the fine-tuning phase. What is the fine-tuning phase? And why do you need it? And what is the relationship between the frequency domain having transferable features and the learning rate value?
- Define your objective in the problem formulation section.

- What is the structure of the encoder G and how did you obtain z_{temp} and z_{freq}? Also, how does the encoder differentiate between domain-specific and domain-invariant features? Also, how does the concatenation of z_{temp} and z_{freq} help with domain adaptation?

- why a hierarchical structure would be more effective for generalizing across domains?
- What is e_h?
- What is “sg” operator in Eq. 4? It should be mentioned directly after the equation.
- The main contribution here is relying on the frequency features to improve the adaptation/transferability between source and target domains. However, the extraction of these features is not clear and adopting them for adaptation is also not clear.

---

### Official Review · Reviewer_Zd5P · 2024-11-04

**Soundness:** 3
**Presentation:** 2
**Contribution:** 2
**Rating:** 5
**Confidence:** 4

**Summary:**

This paper presents TidalFlow, a novel framework that aims to advance time-series unsupervised domain adaptation by leveraging both temporal and frequency domain features. The authors propose that temporal domain features offer superior discriminability while frequency domain features provide better transferability, leading to a complementary integration approach. The method employs a dual-backbone architecture processing temporal and frequency features separately, combined through a voting mechanism. The authors report significant performance improvements, achieving approximately 10% gains across five benchmark datasets

**Strengths:**

- Novel insight into the complementary roles of temporal vs frequency domains
- Proposing a voting mechanism for feature fusion, is quite promising
- Strong empirical results with ~10% improvement on benchmarks

**Weaknesses:**

1- Core Claims Validation:
The fundamental claims about temporal features' discriminability and frequency features' transferability lack rigorous validation. Specific experiments should be designed to quantitatively measure these properties. The authors need to provide eithor theoretical justification or experimental evidence to support these central claims. Without such validation, the paper's foundational assumptions remain questionable.

2- Literature Review and Positioning:
The authors' claim that time-frequency approaches in domain adaptation are underexplored is inaccurate. Several significant works, including FECAM and the Adaptive temporal-frequency network, have made substantial contributions in this area. The paper needs to acknowledge these works and clearly articulate how TidalFlow advances beyond existing approaches.

3- Evaluation Methodology and Confounding Factors
The dual backbone architecture introduces potential confounding factors in performance evaluation. The paper lacks crucial baseline comparisons, particularly source-only performance for both backbones. This makes it difficult to isolate the contribution of the proposed method from architectural advantages.

4- Feature Fusion Analysis
The voting mechanism's superior performance requires better justification. The paper should explore why voting outperforms individual components and compare it with alternative fusion strategies like averaging or summing the time and frequency features. The current analysis doesn't provide sufficient insight into why voting contributes more than the individual time and frequency components.

5- Limited Ablation Studies:
The ablation studies are conducted on only one dataset, which is insufficient for drawing conclusive insights. The authors should extend these studies across multiple datasets to demonstrate the consistency and robustness of their findings. This limitation raises questions about the generalizability of the reported results.

**Questions:**

Nil

---

### Comment · Area_Chair_rF3r · 2024-11-26
**Encouragement to Actively Participate in the Discussion Phase**

Dear Reviewers,

Thank you for your valuable contributions to the review process so far. As we enter the discussion phase, I encourage you to actively engage with the authors and your fellow reviewers. This is a critical opportunity to clarify any open questions, address potential misunderstandings, and ensure that all perspectives are thoroughly considered.

Your thoughtful input during this stage is greatly appreciated and is essential for maintaining the rigor and fairness of the review process.

Thank you for your efforts and dedication.

---

### Meta-Review · Area_Chair_rF3r · 2024-12-19

**Metareview:**

(a) Summary of Scientific Claims and Findings
The paper introduces TidalFlow, a framework for time-series unsupervised domain adaptation (UDA), which leverages the complementary properties of temporal and frequency domain features. The key claims are:
Temporal domain features are more discriminative but domain-sensitive.
Frequency domain features are more domain-invariant but less discriminative.
TidalFlow integrates these features using a dual-backbone architecture, a hierarchical embedding table, and a voting mechanism for feature fusion.
The authors report significant improvements (~10%) over state-of-the-art UDA methods across five benchmark datasets, emphasizing the potential of combining temporal and frequency domain information.

(b) Strengths of the Paper
Novel Perspective: The work revisits the underexplored roles of temporal and frequency domains in UDA, providing valuable insights into their complementary strengths.
Empirical Validation: Results demonstrate improved performance on multiple benchmark datasets, with additional analysis supporting the complementary nature of temporal and frequency domains.
Practical Implementation: The method avoids reliance on source data during adaptation, aligning with the source-free UDA paradigm.
Clear Writing: The paper is generally well-organized, making the proposed framework and experimental results easy to follow.

(c) Weaknesses of the Paper
Lack of Theoretical Depth:
The claims about temporal and frequency domain characteristics are insufficiently validated. No rigorous theoretical or experimental evidence substantiates why these domains exhibit the stated properties.
Core components like the voting mechanism and dissimilarity loss lack detailed theoretical justification.
Inadequate Comparisons and Related Work:
The paper underrepresents related works on time-frequency domain adaptation (e.g., FECAM, Adaptive Temporal-Frequency Network). The distinction between TidalFlow and existing methods remains unclear.
Missing baseline evaluations against state-of-the-art source-free UDA approaches (e.g., SHOT, MAPU) weaken the empirical claims.
Evaluation Limitations:
The datasets used are small-scale and focus narrowly on human activity and sleep analysis, limiting the generalizability of the findings to diverse, large-scale time-series tasks.
Ablation studies and sensitivity analyses are incomplete and primarily confined to a single dataset.
Computational Concerns:
The dual-backbone architecture and hierarchical embedding table introduce potential computational overhead, but no detailed analysis of scalability or computational efficiency is provided.

(d) Reasons for Rejection
Insufficient Validation of Core Claims: The foundational assumptions about temporal and frequency domain properties lack rigorous validation, undermining the scientific contribution of the paper.
Limited Scope and Impact: The evaluation’s narrow focus and lack of diverse datasets reduce the work’s generalizability and applicability to real-world UDA scenarios.
Missed Opportunities in Related Work: The absence of a comprehensive discussion on related methods and insufficient baseline comparisons diminish the contribution.
Unaddressed Reviewer Concerns: The authors' rebuttal improved clarity but failed to resolve critical issues like theoretical depth, broader evaluations, and computational overhead.

While TidalFlow presents an interesting perspective on integrating temporal and frequency domains for UDA, the lack of rigorous validation, broader applicability, and comprehensive comparisons prevent it from meeting ICLR’s standards.

**Additional Comments On Reviewer Discussion:**

Points Raised by Reviewers and Author Responses

Concern: Reviewers questioned the claims that temporal features are more discriminative and frequency features are more domain-invariant. They argued these statements lack rigorous experimental or theoretical validation.
Author Response: The authors added supplementary results comparing feature distributions and confusion matrices for temporal and frequency domains, citing prior studies to support their claims. However, no new theoretical insights or quantitative analyses were provided.
Evaluation: While the additional results provided some empirical support, they did not adequately validate the core claims, leaving the foundational assumptions underexplored.

Concern: Reviewers noted that the paper underrepresents related work on time-frequency domain adaptation (e.g., FECAM, Adaptive Temporal-Frequency Network). Missing baseline comparisons with state-of-the-art UDA methods like SHOT and MAPU were also highlighted.
Author Response: The authors updated the related work section and clarified distinctions between TidalFlow and other methods. They also committed to including comparisons with SHOT and MAPU in future work but did not provide these during the rebuttal.
Evaluation: The updates improved positioning within the literature, but the lack of key baseline comparisons remained a critical gap, reducing confidence in the empirical claims.

Concern: The reviewers raised concerns about the narrow scope of datasets, noting that the benchmarks used are small-scale and limited to specific domains.
Author Response: The authors added a preliminary evaluation on additional datasets (e.g., MFD and Sleep-EDF) and expanded the ablation studies. However, these datasets were still relatively small-scale, and the new results were not comprehensive.
Evaluation: While the added datasets were appreciated, they did not sufficiently address concerns about the method’s generalizability to diverse, large-scale time-series data.

Concern: The reviewers questioned why the voting mechanism and dissimilarity loss led to performance improvements. They also requested comparisons with alternative feature fusion strategies.
Author Response: The authors added sensitivity analyses and compared voting with alternative strategies (e.g., averaging, summing). They argued that voting mitigates incorrect embedding updates during adaptation and provided empirical results supporting its effectiveness.
Evaluation: While the empirical results clarified the benefits of voting, the underlying theoretical explanation remained shallow, leaving the robustness of the design open to question.

Concern: Reviewers asked for an analysis of the computational cost of TidalFlow, especially given its dual-backbone architecture and hierarchical embedding table.
Author Response: The authors included a comparison of training time and memory usage on a single dataset, showing TidalFlow’s overhead was moderate compared to baselines. However, no analysis of scalability to larger datasets was provided.
Evaluation: The added results were helpful but incomplete, leaving concerns about scalability unresolved.

Despite the authors' efforts during the rebuttal period to address reviewer concerns, key issues remained unresolved. The core claims about temporal and frequency domain properties were not rigorously validated, and the evaluation scope was limited to small-scale datasets, reducing the work's generalizability. Additionally, the lack of baseline comparisons with state-of-the-art methods and insufficient theoretical justification for key components (e.g., voting mechanism, dissimilarity loss) weakened the paper's contribution. While improvements in clarity and supplementary results were appreciated, the paper does not meet the standards of novelty, rigor, and broad applicability required for ICLR, leading to a recommendation for rejection.

---

### Decision · Program_Chairs · 2025-01-22

Reject